# Enhancing Offline-to-online Reinforcement Learning by Adaptive Experience Aligned Diffusion Sampling

## Abstract

Pretraining models on diverse prior data and fine-tuning them on domain-specific tasks is an efficient training paradigm to obtain promising performance on scenarios with limited data or interaction. In the context of reinforcement learning (RL), such a paradigm is named offline-to-online (O2O) RL, where the pretrained agent needs to revise and improve the offline pretrained policy based on its own experience in the online environment. Although prior works in the literature have proven the efficiency of fine-tuning the offline-pretrained agent without offline data, they often require additional designs to overcome the unstable online fine-tuning induced by the discrepancy between the offline and online data. Moreover, existing works demonstrate that introducing offline data when training an online agent from scratch is sample-efficient. Therefore, reusing the knowledge from the offline data properly should be favorable to O2O RL. In this paper, we introduce **A**daptive **D**ata **A**ligned **D**iffusion **S**ampling (AD2S), attempting to accelerate the O2O RL fine-tuning from a perspective of data generation. Our method comprises three key components: distance-based experience alignment, curiosity-driven data prioritization, and data regeneration with amplified guidance. AD2S is a plug-in approach and can be combined with existing methods in the offline-to-online RL setting. By implementing AD2S to off-the-shelf methods, Cal-QL, empirical results indicate improvement in commonly studied datasets.

## 1 Introduction

Reinforcement learning (RL) has demonstrated exceptional performance across diverse decision-making and reasoning tasks (DeepSeek-AI et al., 2025; Wang et al., 2018; Zhao et al., 2018; Ling et al., 2024; Lai et al., 2025; Peng et al., 2020; Liu et al., 2025a). However, when implementing the RL paradigm in real-world applications, practitioners often confront a critical challenge: the prohibitive costs and safety risks associated with massive online interaction in safety-critical domains such as autonomous driving or healthcare robotics. This fundamental constraint has catalyzed the development of an efficient learning framework where agents are first pre-trained on comprehensive historical datasets and then fine-tuned on targeted environments — an approach now formally named as offline-to-online (O2O) RL (Liu et al., 2024; Zhang et al., 2024; Zhou et al., 2024).

Nevertheless, when deploying the offline pretrained agent to the online environment, two critical challenges emerge, resulting in unstable online Q-learning: (1) Due to the penalization of out-of-distribution (OOD) actions during offline training, the inherent pessimism of offline-pretrained Q networks to OOD actions often leads to overly conservative policy updates; (2) The non-negligible distributional discrepancy between the offline dataset and the online replay buffer induces catastrophic forgetting and suboptimal convergence. Existing methods replay offline data and introduce specific learning paradigms to address the significant distribution gaps between offline datasets and online samples during offline-to-online fine-tuning. For example, they may consider aligning the policy to be consistent with the behavior policies in both offline and online datasets (Nair et al., 2020), leveraging the capacity of the model ensemble to balance the agent performance and training stability (Lee et al., 2021; Zhao et al., 2022), introducing regularization on Q networks (Zhang et al., 2024), constructing a unified learning paradigm for sequential modeling (Zheng et al., 2022), or introducing policy expansion (Zhang et al., 2023; Uchendu et al., 2023).

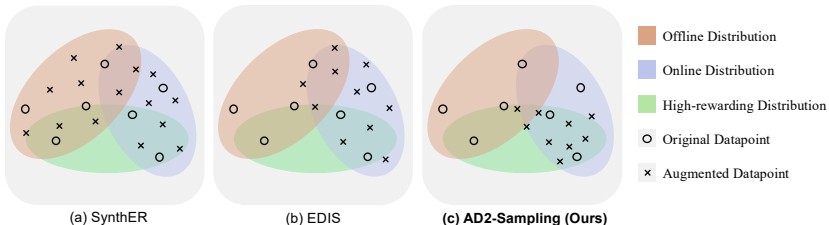

Figure 1: Comparison of previous diffusion-based data generator in O2O RL (Lu et al., 2023c; Liu et al., 2024) and AD2S. AD2S incorporates adaptive data reuse and diffusion-based regeneration, pushing the data towards high-rewarding, under-explored regions for sample-efficient O2O RL.

Recent works have established that agent fine-tuning without offline data replay consistently outperforms the aforementioned fine-tuning methods (Zhou et al., 2024; Liu et al., 2024). However, state-of-the-art solutions typically demand computationally intensive operations, such as high-frequency Q-net updates with model ensembles (Zhou et al., 2024; Zhang et al., 2024) or energy-guided diffusion sampling (Liu et al., 2024). In this paper, we attempt to reuse the key knowledge from offline data and accelerate the online fine-tuning from a data generation perspective. Our key insight is, *Can we adaptively generate synthetic data that is beneficial to O2O RL fine-tuning?*

To verify our insight and accelerate the online fine-tuning phase in O2O RL, we introduce **A**daptive **D**ata **A**ligned **D**iffusion **S**ampling (AD2S or AD$^2$S). Our approach comprises three key components: *distance-based experience alignment*, *curiosity-driven data prioritization*, and *data regeneration with amplified guidance*. Firstly, AD2S aligns offline data that is close to the online experiences, facilitating stable Q-learning through effective dataset reuse. Secondly, AD2S incorporates a curiosity-driven mechanism to assess buffer novelty, adaptively identifying high-novelty transitions (see Figure 1). Thirdly, AD2S utilizes partial noising on the pre-aligned data and conditions the diffusion model to regenerate synthetic data with amplified guidance. These mechanisms enable AD2S to replay the near on-policy, high-novelty experience from the seen data and ensure sufficient online exploration.

Overall, our key contributions are: (1) We introduce AD2S, a simple yet effective framework incorporating a data alignment mechanism and a diffusion model to adaptively generate high-fidelity training data. (2) AD2S replays historical samples based on advantage-weighted relative metrics and regenerates the aligned data towards high-rewarding and under-explored regions for sample-efficient online fine-tuning in O2O RL. (3) Through extensive experiments on popular O2O tasks, empirical results demonstrate that AD2S achieves superior performance compared to previous SOTA methods without any modifications to the backbone algorithms. (4) We assess the synthetic dataset generated by AD2S with data quality metrics, proving its alignment with the objective of O2O RL. These findings validate AD2S as an effective paradigm for accelerating online fine-tuning in O2O RL.

## 2 PRELIMINARIES

### 2.1 REINFORCEMENT LEARNING

Reinforcement Learning (RL) is formulated as a Markov decision process (MDP) described by the tuple $(\mathcal{S}, \mathcal{A}, \mathcal{T}, \mathcal{R}, \gamma)$, consisting of state space $\mathcal{S}$, action space $\mathcal{A}$, transition function $\mathcal{T} : \mathcal{S} \times \mathcal{A} \to \mathcal{S}$, reward function $\mathcal{S} \times \mathcal{A} \times \mathcal{S} \to \mathcal{R}$, and discount factor $\gamma \in [0, 1)$ (Sutton & Barto, 1998). At each timestep $t$, the agent selects an action $a_t$ according to the policy $\pi$ conditioned on the state $s_t$. Consequently, the agent receives a reward $r_t$ for the action $a_t$ taken in the state $s_t$, and the environment transforms to the next state $s_{t+1} \sim \mathcal{T}(\cdot|s_t, a_t)$. The goal of RL is to learn a policy $\pi^*$, which maximizes expected discounted return, $J(\pi) = E_\pi[\sum_{t=0}^{\infty} \gamma^t r_t]$. Generally, there are two learning paradigms of RL: online RL, where the agent can learn from interacting with the environment; and offline RL, where the agent can only learn from a fixed dataset $\mathcal{D}^{\text{off}} = \{(s, a, r, s')\}$, which has been collected using an unknown behavior policy $\pi_\beta$.

**Offline-to-online (O2O) Reinforcement Learning.** O2O RL bridges offline pretraining with online fine-tuning, aiming to leverage historical data to train a near-optimal policy under limited online

interaction. The agent is first pretrained on a fixed dataset $\mathcal{D}^{\mathrm{off}} = \{(s, a, r, s')\}$, then explores in the online environment to recover from suboptimal behaviors and refine its policy.

## 2.2 DIFFUSION MODELS

**Score-based diffusion models.** Diffusion models (Ho et al., 2020; Karras et al., 2022) are a class of generative models inspired by non-equilibrium thermodynamics. Consider a data distribution $p(\mathbf{x})$ with standard deviation $\sigma_{\mathrm{data}}$, diffusion models gradually add i.i.d. Gaussian noise of standard deviation $\sigma$ on the base distribution from time 0 to $K$ and obtain noised distributions $p(\mathbf{x}; \sigma)$. The forward noising process is defined by a sequence of noised distributions following a fixed noise schedule $\sigma_0 = \sigma_{\max} > \sigma_1 > \cdots > \sigma_N = 0$ so that at each noise level, $\mathbf{x}^k \sim p(\mathbf{x}^k; \sigma_k)$. When $\sigma_{\max} \gg \sigma_{\mathrm{data}}$, the final noised distribution $p(\mathbf{x}^K; \sigma_{\max})$ is essentially indistinguishable from random noise. The diffusion model is trained to iteratively denoise samples from a Gaussian distribution and ultimately recover the target distribution, which is formally named the reverse process. Karras et al. (Karras et al., 2022) consider this process as a probability-flow ODE and formulate as below:

$$\mathrm{d}\mathbf{x} = -\dot{\sigma}(k)\sigma(k)\nabla_{\mathbf{x}} \log p(\mathbf{x}; \sigma(k))\mathrm{d}k, \tag{1}$$

where $\nabla_{\mathbf{x}} \log p(\mathbf{x}; \sigma(k))$ denotes the score function, which points towards the data for a given noise level, and the dot indicates a time derivative. The denoiser $G_\theta(\mathbf{x}_t; \sigma)$ is trained on an L2 denoising minimization objective:

$$\mathcal{L}(G_\theta; \sigma) = \mathbb{E}_{\mathbf{x} \sim p, \epsilon \sim \mathcal{N}(0, \sigma^2 I)} \|G_\theta(\mathbf{x} + \epsilon; \sigma) - \mathbf{x}\|_2^2, \tag{2}$$

and the score can be calculated by $\nabla_{\mathbf{x}} \log p(\mathbf{x}; \sigma) = (G_\theta(\mathbf{x}; \sigma) - \mathbf{x})/\sigma^2$. In this paper, we sample data via solving Eq. 1 with the learned denoising network.

**Conditional score-based diffusion model.** For additional controllability, diffusion models naturally enable conditioning on some signal $y$ (Dhariwal & Nichol, 2021; Ho & Salimans, 2022). Classifier-free guidance (CFG) (Ho & Salimans, 2022) is a common post-training technique that further promotes sample fidelity to the condition $y$ in exchange for more complete mode coverage. The guidance distribution $\tilde{p}_\theta$ is interpreted as $\tilde{p}_\theta(\mathbf{x}|y) \propto p_\theta(\mathbf{x}|y) \cdot p_\theta(y|\mathbf{x})^\eta$. Subsequently, considering the equivalence relationship between score matching and the denoising process $\nabla_{\mathbf{x}} \log p_\theta(\mathbf{x}|y) \propto \epsilon_\theta(\mathbf{x}, y)$ with the implicit classifier $p_\theta(y|\mathbf{x}) \propto p_\theta(\mathbf{x}|y)/p_\theta(\mathbf{x})$, the CFG score $\tilde{\epsilon}_\theta$ can be formed as:

$$\tilde{\epsilon}_\theta(\mathbf{x}^k|y) = (\eta + 1) \cdot \epsilon_\theta(\mathbf{x}^k, y) - \eta \cdot \epsilon_\theta(\mathbf{x}^k, \varnothing), \tag{3}$$

where $\eta$ is a hyparameter called the *guidance scale*. The training objective of the CFG is to concurrently train the conditional and unconditional score functions as follows, where $\lambda$ is the dropout rate of condition $y$:

$$\mathcal{L}(\theta) = \mathbb{E}_{k, \epsilon, \mathbf{x}^0 \sim \mathcal{D}, y, \lambda \sim \mathrm{Bernoulli}(\lambda)} \left[ \|\epsilon - \epsilon_\theta(\mathbf{x}^k, (1 - \lambda)y + \lambda\varnothing)\|^2 \right]. \tag{4}$$

## 3 METHOD

In this section, we introduce Adaptive Data Alignment Diffusion Sampling (AD2S). At its core, AD2S accelerates online fine-tuning in O2O RL through three key mechanisms: (1) distance-based data alignment by reusing near on-policy data from the offline and online samples, (2) curiosity-driven data prioritization from aligned data to enhance online exploration, and (3) amplified condition guided diffusion synthesizer to push the data towards high-rewarding and under-explored regions. We first provide motivation for the AD2S, and concretize how it can be instantiated. Next, we elaborate on the data alignment and generation pipeline. Finally, we present the overall training procedure.

### 3.1 MOTIVATIONS

The deployment of the offline pretrained agent in online environments presents two fundamental challenges that hinder effective policy improvement. Firstly, the significant distribution shift between offline datasets and online collected samples induces an unstable Q-learning procedure, leading to catastrophic forgetting of the pretrained Q-function. Moreover, almost all offline pretrained Q-functions are overly pessimistic about OOD actions, as they attempt to penalize these actions during offline training, which creates exploration barriers that prevent effective online fine-tuning.

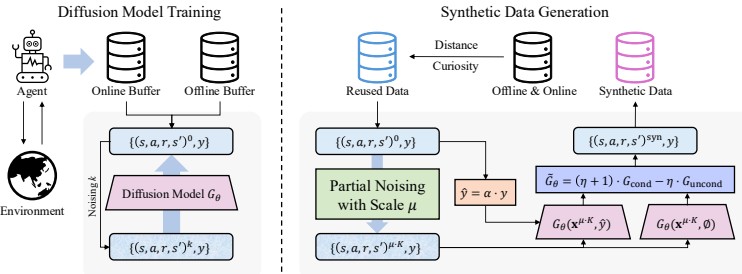

Figure 2: Overall framework of the AD2S. The diffusion model $G_\theta$ is trained on seen data $\mathcal{D}^{\text{off}} \cup \mathcal{D}^{\text{on}}$. AD2S conducts key improvements during data generation: (1) adaptively aligning seen data to near on-policy, high-novelty data based on Eqs. 5 and 9, (2) partial nosing on aligned data via diffusion forward process, and (3) leverage the diffusion model to regenerate high-rewarding, under-explored synthetic data by amplified condition guidance.

To enable a stable online Q-function fine-tuning, we introduce a distance-based metric in Eq. 5 to identify near on-policy samples $\mathcal{D}^{\text{aligned}}$ from the offline data and online experience. The aligned data helps the pretrained Q-function to mitigate the distribution discrepancy and avoid catastrophic forgetting. Regarding the inherent pessimism in Q-functions, we introduce a curiosity-driven mechanism in Eq. 9 to prioritize and select high-novelty data from the distance-based aligned data $\mathcal{D}^{\text{aligned}}$ as reused data $\mathcal{D}^{\text{re}}$. Reusing the high-novelty data enables the agent to calibrate the offline pretrained Q network and enhance online exploration, thus accelerating the online fine-tuning. Moreover, we introduce a diffusion-based generator to enrich the reused data. Generating the reused data parametrically not only empowers the extrapolation of Q-functions but also interpolates the data distribution to more impoverished, high-rewarding data regions.

Figure 2 gives a brief illustration of AD2S. We first train the conditional diffusion model $G_\theta$ on offline and online collected samples. The advantage-weighted curiosity function in Eq. 9 defines the condition $y$. Then, we select the data $\mathcal{D}^{\text{re}}$ for generation by leveraging the distance and curiosity metric. We use the diffusion model to regenerate them on amplified signals conditionally. This enables targeted density of the buffer distribution to high-rewarding, under-explored regions.

## 3.2 DISTANCE-BASED METRIC FOR DATA ALIGNMENT

Adaptively replaying near on-policy experience from offline data can stabilize the Q-learning and significantly enhance sample efficiency in online learning (Liu et al., 2025b; Ball et al., 2023). In O2O RL, we aim to dynamically balance the reuse of offline samples with online experience, therefore mitigating distributional shift while enriching the online replay buffer and preserving policy improvement potential. We use the advantage-weighted priority $u$ to represent the distance between online and offline samples,

$$u = u(s, a, r, s') = w(s, a, r, s') \cdot \exp(\beta \cdot A(s, a)) \tag{5}$$

where $A(s, a)$ is the advantage term, which indicates the potential of the transition for policy improvement, $\beta > 0$ represents a temperature value, and $w(\cdot)$ denotes the density ratio that measures the relative distance of the transition which can be formulated as below,

$$w(s, a, r, s') := d^{\text{on}}(s, a, r, s') / d^{\text{off}}(s, a, r, s') \tag{6}$$

for a given transition $(s, a, r, s')$, where $d^{\text{on}}(\cdot)$ denotes the transition distribution of online samples in the online buffer $\mathcal{D}^{\text{on}}$ and the $d^{\text{off}}(\cdot)$ represents the offline samples in the offline dataset $\mathcal{D}^{\text{off}}$. This distance metric provides an efficient way to identify near on-policy transitions from online and offline samples. To estimate the proposed density ratio, we approximate $w(\cdot)$ by training a neural network $w_\psi(\cdot)$ and use the variational representation of $f$-divergences (Nguyen et al., 2007). Consider $P$ and $M$ as probability measures on a measurable space $\mathcal{X}$, with $P$ being absolutely continuous w.r.t $M$. We define the function $f(y) := y \log \frac{2y}{y+1} + \log \frac{2}{y+1}$. Then we could define the Jensen-Shannon (JS) divergence as $D_{JS}(P\|M) = \int_{\mathcal{X}} f(dP(\mathbf{x})/dM(\mathbf{x}))dM(\mathbf{x})$. Therefore, the density ratio $\frac{dP}{dM}$ could be formed by $w_\psi(\mathbf{x})$ and be estimated by maximizing the lower bound of $D_{JS}(P\|M)$,

$$\mathcal{L}_{\text{DR}}(\psi) = \mathbb{E}_{\mathbf{x} \sim P}\left[f'(w_\psi(\mathbf{x}))\right] - \mathbb{E}_{\mathbf{x} \sim M}\left[f^*(f'(w_\psi(\mathbf{x})))\right], \tag{7}$$

where $w_\psi(\mathbf{x}) \geq 0$ is represented by a neural network, $f'$ is the derivative of $f$ and $f^*$ indicates the convex conjugate of $f$. We sample from $\mathcal{D}^{\mathrm{on}}$ for $\mathbf{x} \sim P$ and from $\mathcal{D}^{\mathrm{off}}$ for $\mathbf{x} \sim M$.

The traditional way of advantage estimation $A(s,a)$ is to train an advantage function $A(s,a) = Q(s,a) - V(s)$. However, the distribution discrepancy between online and offline samples leads to inaccurate estimation during the early online fine-tuning phase (Zhang et al., 2024; Zhou et al., 2024). To overcome this issue, we introduce the statistical-based relative advantage estimation, which can be formulated as follows,

$$A(s,a) = (r(s,a) - r_{\mathrm{mean}})/r_{\mathrm{std}}, \tag{8}$$

where $r_{\mathrm{mean}}$ and $r_{\mathrm{std}}$ are calculated from online and offline samples. The relative advantage estimation provides a calibrated and stable advantage estimation, which proves particularly crucial during the early stage of the online fine-tuning phase in O2O RL. In this way, by measuring $u$ proposed in Eq. 5 with parametric density ratio $w_\psi$, we could construct aligned data $\mathcal{D}^{\mathrm{aligned}}$ for curiosity-driven prioritization and diffusion-based regeneration.

### 3.3 CURIOSITY-DRIVEN DATA ALIGNMENT AND DIFFUSION-BASED DATA GENERATION

Implementing curiosity-driven data generation is an effective way to overcome the inherent pessimism in offline-pretrained Q-functions by enhancing online exploration. In AD2S, we use a forward dynamics model $g$ as the curiosity estimator to construct the reused buffer $\mathcal{D}^{\mathrm{re}}$ from aligned data $\mathcal{D}^{\mathrm{aligned}}$, and a diffusion model to regenerate data towards high-rewarding, under-explored regions. To train the forward dynamics model $g_\phi(s,a)$, we use the data $\mathbf{x}$ sampled from offline and online samples $\mathcal{D}^{\mathrm{off}} \cup \mathcal{D}^{\mathrm{on}}$ and minimize the transition error between the real and predicted next state,

$$e(s,a,s',r) = \|s' - \hat{s}'\|^2 \quad \text{where } \hat{s}' = g_\phi(s,a). \tag{9}$$

We also integrate the relative advantage metric (Eq. 8) into the error measurement $y(\mathbf{x}) = \exp(\beta \cdot A(s,a)) \cdot e(s,a,r,s')$ to prioritize high potential reward in under-explored regions. We utilize the advantage-weighted metric to perform curiosity estimation for seen samples. Therefore, based on the aforementioned estimations, our framework could adaptively identify near on-policy data with high-novelty and construct the reused data $\mathcal{D}^{\mathrm{re}}$.

For the diffusion model training, we randomly sample data $\mathbf{x}$ from offline and online buffers $\mathcal{D}^{\mathrm{off}} \cup \mathcal{D}^{on}$ and require the diffusion model $G_\theta(\mathbf{x}|y(\mathbf{x}))$ to approximate the conditional distribution $p(\mathbf{x}|y(\mathbf{x}))$, where the condition signal is also defined by the advantage-weighted curiosity: $y(\mathbf{x}) = \exp(\beta \cdot A(s,a)) \cdot e(s,a,r,s')$. Training on $\mathcal{D} = \mathcal{D}^{\mathrm{off}} \cup \mathcal{D}^{on}$ enables $G_\theta$ to learn the whole conditioned distribution. Considering the equivalence relationship between score matching and the denoising process described in Section 2.2, The objective for updating parameter $\theta$ with the dropout rate $\lambda$ of condition $y$ is below,

$$\theta^* \leftarrow \arg\min_\theta \mathbb{E}_{k,\epsilon,\mathbf{x}\sim\mathcal{D},\lambda\sim\mathrm{Bernoulli}(\lambda)} \left[ \|\epsilon - \epsilon_\theta(\mathbf{x}^k|(1-\lambda)y + \lambda\varnothing)\|^2 \right]. \tag{10}$$

To push the aligned data $\mathcal{D}^{\mathrm{aligned}}$ towards high-rewarding, under-explored regions, we add partial noise to these data and use the conditional diffusion model to regenerate them with amplified guidance (Lee et al., 2024; Huang et al., 2024b). Specifically, let $\mathbf{x} = \mathbf{x}^0 \sim \mathcal{D}^{\mathrm{re}}$ denotes the original aligned data, and $k \in [1, K]$ denotes the diffusion timestep. Our method first injects controlled noise into $\mathbf{x}^0$ through a truncated forward process $\mathbf{x}^{\mu \cdot K} \sim \mathcal{N}(\mathbf{x}; \mathbf{x}^0, \sigma(\mu \cdot K)^2 I)$, where the exploration parameter $\mu(0 < \frac{k}{K} \leq 1)$ governs the noise intensity, trading off between preserving original transition features $(\mu \to 0)$ and enabling novel sample generation $(\mu \to 1)$. Crucially, we amplify the guidance signal $y$ of the state $s$ during the reverse diffusion process by $\hat{y} = \alpha \cdot y$ where $\alpha > 1$ to enhance the novelty of the regenerated transitions. This denoising process in each step $k$ is formally defined by,

$$\tilde{\epsilon}_\theta(\tilde{\mathbf{x}}^k|y) = (\eta + 1) \cdot \epsilon_\theta(\tilde{\mathbf{x}}^k, \hat{y}) - \eta \cdot \epsilon_\theta(\mathbf{x}^k, \varnothing), \tag{11}$$

where $\tilde{\mathbf{x}}$ is the regenerated samples guided by amplified guidance, $\eta$ controls the scale of the guidance. This generation mechanism promotes the synthetic data to retain fidelity to task-relevant patterns while extrapolating toward under-explored, high-rewarding regions of the transition space.

### 3.4 FRAMEWORK SUMMARY

Finally, we provide a concrete overview of our framework in Algorithm 1. After being pretrained on the offline dataset $\mathcal{D}^{\mathrm{off}}$, the agent interacts with the environment, collecting a stream of real data and

---

**Algorithm 1** Overview of AD2S framework.

---

1: **Require:** synthetic ratio $p$, density-based alignment ratio $p_{DR}$, curiosity-driven alignment ratio $p_{Curi}$, conditional guidance scale $\eta$, amplified scale $\alpha$, offline pretrained agent $\pi$
2: Initialize $w_\psi$, $G_\theta$, offline buffer $D^{off}$, online buffer $D^{on}$, dynamics model $g_\phi$
3: **while** in *online training phase* **do**
4:     Collect transitions $(s, a, r, s')$ with $\pi$ in the environment and add to $\mathcal{D}^{on}$
5:     Update $w_\psi$ and $g_\phi$ using $\mathcal{D}^{on}$ via Eqs. 7 and 9
6:     **if** steps meets $G_\theta$ update frequency **then**
7:         Update $G_\theta$ using samples from $\mathcal{D}^{on} \cup \mathcal{D}^{off}$ via Eq. 4
8:         Construct aligned buffer $\mathcal{D}^{re}$ by calculating $u$ and $y$ using $w_\psi$, $g_\phi$, $p_{DR}$, and $p_{Curi}$
9:         Conditionally generate synthetic data by $G_\theta$ with data from $\mathcal{D}^{re}$ and amplified condition using Eq. 11
10:        Construct synthetic buffer $\mathcal{D}^{syn}$ using data generated by $G_\theta$
11:    Train $\pi$ on samples from $\mathcal{D}^{on} \cup \mathcal{D}^{syn}$ mixed with ratio $p$
12: **end while**

---

constructing the online replay buffer $\mathcal{D}^{on}$. We also update the parametric density ratio network $w_\psi$ and forward dynamics model $g$ using samples from $\mathcal{D}^{off}$ and $\mathcal{D}^{on}$, via the loss function given by Eqs. 7 and 9. The conditional diffusion model $G_\theta$ is trained on the mixed dataset $\mathcal{D}^{on} \cup \mathcal{D}^{off}$ using Eq. 10. For the data generation, we first build the $\mathcal{D}^{aligned}$ by selecting data from offline and online data with the highest metric for $u(s, a, r, s')$ based on ratio $p_{DR}$, then we construct $\mathcal{D}^{re}$ by measuring the highest curiosity $y(\mathbf{x})$ on $\mathcal{D}^{aligned}$ and select data with ratio $p_{Curi}$. Then we utilize the conditional diffusion model to generate under-explored, high-rewarding synthetic data $\mathcal{D}^{syn}$ using Eq. 11 with data from $\mathcal{D}^{re}$ and amplified condition $\hat{y}$. The $\mathcal{D}^{syn}$ is ultimately used for online fine-tuning. The sampling ratios $p_{DR}$ and $p_{Curi}$ are predefined hyperparameters, a smaller $p_{DR}$ or $p_{Curi}$ corresponds to a more extreme data alignment or curiosity-driven exploration strategy, respectively. We also present the empirical results on how to choose $p_{DR}$ and $p_{Curi}$ in Section C.1.

## 4 EXPERIMENTS

In this section, we conduct extensive experiments across commonly studied benchmarks to answer the following questions: (1) How much performance gain does AD2S exhibit across various tasks? (2) What are the underlying mechanisms by which AD2S brings about performance gains? (3) Does AD2S synthesize high-fidelity data?

### 4.1 EXPERIMENTAL SETUP

**Datasets and environments.**  We evaluate the performance of AD2S on three commonly studied benchmarks from the canonical D4RL dataset (Fu et al., 2020), such as MuJoCo Locomotion and Maze2D. These benchmarks help us to validate AD2S under different scenarios. In all tasks, we allow 200K environment interactions for online fine-tuning, which facilitates direct comparison to existing methods (Liu et al., 2024).

**Baselines.**  We compare AD2S with existing augmentation methods based on diffusion models: (1) **SynthER.** Lu et al. (2023c) unconditionally generates synthetic data based on the diffusion model, which can be deployed on both offline and online stages. Here, we directly implement SynthER during the fine-tuning stage for online data generation. (2) **EDIS.** Liu et al. (2024) leverages an energy model to capture the distribution of online data, regarding it as the classifier-guidance of the diffusion model to generate near on-policy data. (3) **PGR.** Wang et al. (2024) considers multiple relevance functions to prioritize the online data, and utilizes the diffusion model to interpolate the replay distribution to more impoverished data regions.

For all D4RL benchmarks, we implement AD2S and baselines on top of base algorithms Cal-QL (Nakamoto et al., 2023), a state-of-the-art O2O method that effectively calibrates over-conservatism of CQL (Kumar et al., 2020). All methods are pretrained on the offline dataset and fine-tuned on the online environment for 0.2M steps. Implementation details are referred to Appendix A.

Table 1: Normalized average scores on O2O RL tasks over five random seeds. Here we report the best results for SynthER, PGR and AD2S, and use the results from Liu et al. (2024) for Cal-QL and EDIS. We highlight the best scores in **bold**, and underline the AD2S's score close to the best score.

| Dataset | Cal-QL | SynthER | PGR | EDIS | AD2S (Ours) |
|---|---|---|---|---|---|
| hopper-random-v2 | 17.6± 3.1 | 33.1± 1.7 | 51.9±37.7 | 98.1±12.3 | **110.7± 4.3** |
| hopper-medium-replay-v2 | 102.2± 4.6 | 108.8± 1.5 | 102.1± 1.8 | **109.9± 0.8** | 108.9± 2.2 |
| hopper-medium-v2 | 97.6± 1.4 | 106.8± 1.4 | 108.1± 3.0 | 105.0± 4.1 | **109.0± 3.8** |
| hopper-medium-expert-v2 | 107.9± 9.6 | 111.6± 0.6 | 111.8± 1.0 | 109.7± 1.4 | **112.1± 1.4** |
| halfcheetah-random-v2 | 74.8± 3.2 | 60.1± 8.0 | 79.0± 6.8 | 86.3± 1.8 | **90.1± 4.5** |
| halfcheetah-medium-replay-v2 | 76.6± 1.2 | 79.3± 2.0 | 83.1± 2.1 | 86.7± 1.4 | **92.3± 1.9** |
| halfcheetah-medium-v2 | 72.3± 2.1 | 88.2± 1.8 | 84.5± 0.9 | 83.9± 1.0 | **90.8± 4.7** |
| halfcheetah-medium-expert-v2 | 91.0± 0.6 | 86.6± 4.2 | 98.3± 1.3 | **98.6± 0.5** | 93.9± 1.5 |
| walker2d-random-v2 | 15.1± 3.5 | 42.7±29.7 | 66.0±16.9 | 61.6±12.6 | **99.3±11.6** |
| walker2d-medium-replay-v2 | 87.3± 8.5 | 109.7± 3.7 | 118.6± 1.1 | 112.9± 6.4 | **121.0± 7.6** |
| walker2d-medium-v2 | 84.2± 0.3 | 108.4± 2.3 | 114.8± 3.0 | 103.5± 1.8 | **119.4± 1.2** |
| walker2d-medium-expert-v2 | 111.1± 0.6 | 112.6± 1.2 | 122.2± 8.2 | 118.5± 4.0 | **129.1± 4.6** |
| locomotion total | 937.7 | 1047.8 | 1140.6 | 1174.7 | **1276.6** |
| maze2d-umaze-v1 | 51.4±17.7 | **171.3± 5.2** | 157.5± 8.1 | 162.9± 4.7 | 169.6± 4.8 |
| maze2d-medium-v1 | 25.4± 2.2 | 185.1± 5.5 | 179.3±15.5 | 186.4± 5.0 | **188.6± 6.5** |
| maze2d-large-v1 | 3.9± 7.0 | 211.1±14.0 | 163.5±19.8 | 209.3±30.5 | **228.2±15.3** |
| maze2d total | 80.6 | 567.5 | 500.3 | 558.6 | **586.4** |

Table 2: D4RL normalized scores over five random seeds on 3 tasks, with the highest scores highlighted in **bold**. We conduct experiments on MuJoCo locomotion tasks with low data quality to investigate the effectiveness of different components in AD2S.

| Dataset | w/o DA | w/o PN | w/o CG | AD2S (Ours) |
|---|---|---|---|---|
| hopper-random-v2 | 20.7± 8.0 | 27.6± 6.3 | 105.1±11.9 | **110.7± 4.3** |
| halfcheetah-random-v2 | 50.0± 4.5 | 60.3±24.7 | 86.4± 1.4 | **90.1± 4.5** |
| walker2d-random-v2 | 20.6± 3.5 | 68.5±10.6 | 56.6±31.4 | **99.3±11.6** |

## 4.2 MAIN RESULTS

Our results in Table 1 show that AD2S outperforms existing diffusion-based data synthesizers in various tasks, especially for policies trained on low-quality datasets. Compared to SynthER and PGR, AD2S achieves significant improvement on MuJoCo random datasets, indicating that our proposed framework can reduce the impact of low-quality offline datasets on the diffusion-based synthesizer. Meanwhile, our method demonstrates that enriching the high-curiosity data from advantage-weighted near on-policy samples is more sample-efficient than only generating high-curiosity data. Moreover, AD2S obtains further improvement compared to EDIS, indicating the effectiveness of pushing the synthetic data to high-rewarding novel regions. More empirical results are referred to Appendix B.

## 4.3 ABLATION STUDIES

To verify the effectiveness of each component in AD2S, we conduct ablation studies on walker2d-random-v2, hopper-random-v2, and halfcheetah-random-v2 datasets with the following variants of AD2S: (1) without data alignment (w/o DA), (2) without partial noising (w/o PN) (3) without condition guidance (w/o CG). We report the results in Table 2 and below are key findings: (i) Data alignment can lead the synthesizer to generate near on-policy data, further improving the performance. (ii) When considering amplified condition guidance, partial noising constrains the distance between the synthetic and ground-truth samples, thus acquiring a more sample-efficient online fine-tuning. (iii) Unconditional diffusion sampling encounters performance degradation on low-quality data, demonstrating the effectiveness of our proposed conditional diffusion sampling. More ablation studies are referred to Appendix C.

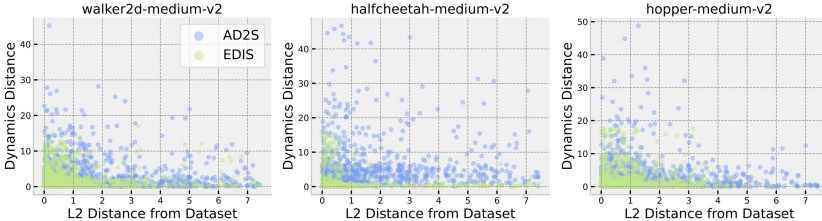

Figure 3: We plot the L2 distance and the dynamic distance under AD2S or EDIS from the online collected data. Compared to EDIS, which is energy model guided diffusion sampling, AD2S can adaptively generate higher novelty data than the existing SOTA method.

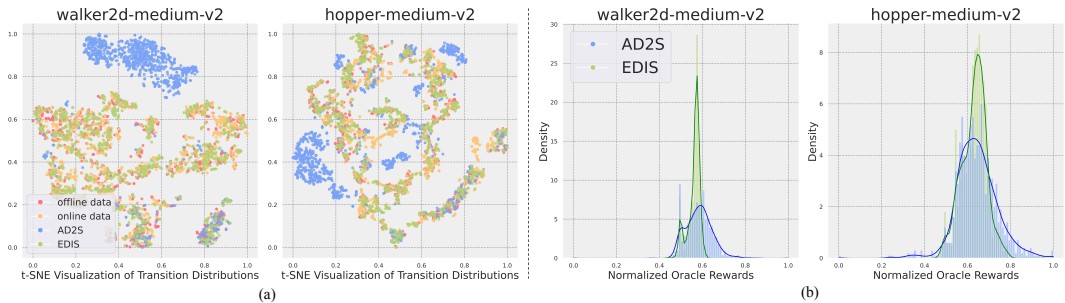

Figure 4: We visualize the transitions of AD2S and EDIS data by t-SNE **(a)**. We also plot the Oracle rewards for them **(b)**. The results demonstrate that AD2S not only generates data with higher curiosity but also pushes the synthetic data towards higher-rewarding regions.

## 4.4 SYNTHETIC DATA ANALYSIS

To provide an intuition into the efficacy of our proposed method, we follow previous works (Lu et al., 2022; 2023c), using the ground-truth simulator to measure the dynamics distance (i.e., MSE error) between AD2S or EDIS (Liu et al., 2024) with the real next state to verify the transition-level curiosity and validity of the synthetic samples. We also measure the curiosity from the perspective of a distance metric by calculating the L2 distance between the synthetic data and the mean of the real datasets. The results are presented in Figure 3. Compared to EDIS, AD2S generates data with a larger L2 distance. Moreover, data generated by AD2S has a larger dynamic distance, which encourages the agent to explore the online environment and refines the knowledge learned from synthetic data (Wang et al., 2024), thus speeding up the fine-tuning process.

To further verify the fidelity of the synthetic samples, we visualize the synthetic data distribution between AD2S and EDIS using t-SNE (van der Maaten & Hinton, 2008) on walker2d-medium and hopper-medium tasks in Figure 4. We also plot the Oracle rewards defined by the simulator. The results demonstrate that AD2S not only generates data with higher curiosity but also pushes the synthetic data towards higher-rewarding regions.

## 4.5 ONE-STEP ADVANTAGE ANALYSIS

The traditional advantage $A(s, a) = Q(s, a) - V(s)$ estimates the additional return from taking action $a$ in state $s$. In contrast, Eq. 8 evaluates transitions relative to the entire buffer, identifying those most critical transitions for training. To validate its efficacy, we approximate the advantage in AD2S using the Q net from the agent itself: $A(s, a) = Q(s, a) - \mathbb{E}[Q(s, \tilde{a})]$, where $\tilde{a}$ denotes sampled random actions. Results on the Walker2d environment (Table 3) demonstrate its effectiveness. We regard that the neural-network-based advantage estimation may introduce instability during early-stage fine-tuning, necessitating additional regularization for Q models, which can lead to training difficulties and increased computational requirements.

Table 3: D4RL normalized scores over five seeds on the Walker2d task with 4 data qualities. We conduct experiments to validate the efficacy of our short-term advantage proposed in Eq. 8.

| Method | random | medium | medium-replay | medium-expert |
|---|---|---|---|---|
| AD2S | **99.3**±**11.6** | **119.4**± **1.2** | **121.0**± **7.6** | **129.1**± **4.6** |
| AD2S (variant) | 12.6± 5.9 | 85.1± 1.8 | 112.5± 7.1 | 110.0± 1.4 |

## 5 RELATED WORK

### 5.1 OFFLINE-TO-ONLINE RL

Offline-to-online RL methods are developed to bridge the high asymptotic performance in online RL and the low exploration cost in offline RL. The learning process focuses on leveraging the offline dataset to pre-train an agent to run online RL as sample-efficiently as possible (Lee et al., 2021; Nair et al., 2020; Liu et al., 2025b; Ball et al., 2023; Tarasov et al., 2023a). The commonly studied paradigms utilize offline pretraining followed by a particularly designed fine-tuning phase, such as policy expansion (Zhang et al., 2023; Uchendu et al., 2023), value function calibration (Nakamoto et al., 2023), Q-ensemble techniques (Lee et al., 2021; Wang et al., 2023a), regularization (Zhang et al., 2024), and constraint methods (Nair et al., 2020; Kostrikov et al., 2022; Li et al., 2023). Although retaining offline data during fine-tuning can tackle the over-conservatism of the agent (Fujimoto et al., 2019; Fujimoto & Gu, 2021; Kumar et al., 2020) and prevent catastrophic forgetting, recent works show that fine-tuning the pretrained agent without offline data achieves a better asymptotic performance. Zhou et al. (2024) proposes a simple but effective way to revise the Q function during fine-tuning. Liu et al. (2024) leverages the capacity of the energy model, guiding the diffusion model to generate near on-policy data for sample-efficient fine-tuning. In this paper, we take the advantages of both sides, integrating a weighted density ratio mechanism to select near on-policy data from historical data and leveraging the conditional diffusion model to generate high-fidelity data.

### 5.2 DIFFUSION MODELS AS DATA GENERATOR IN RL

Diffusion models have demonstrated outstanding capabilities in modeling complex distributions (Ho et al., 2020; Saharia et al., 2022; Nichol et al., 2022; Nichol & Dhariwal, 2021; Song et al., 2023). Recent works have employed diffusion models in offline RL for action execution, with extensions to multi-task settings and the alignment of human preferences (Janner et al., 2022; Ajay et al., 2023; Ren et al., 2024; Wang et al., 2023b; Lu et al., 2023a; He et al., 2023b; Jain & Ravanbakhsh, 2024; Mao et al., 2024; He et al., 2023a; Dong et al., 2024). Besides that, another idea is to utilize the capabilities to generate synthetic data in both offline and online RL (Lu et al., 2023c; Lee et al., 2024; Li et al., 2024; Jackson et al., 2024; Liu et al., 2024). GTA (Lee et al., 2024) and TD (Huang et al., 2024a) introduce partial noising on some trajectories, treating the diffusion model as an optimizer to generate high-fidelity trajectories. Moreover, previous works also leverage the capabilities for trajectory stitching (Ghugare et al., 2024), generating the trajectories that do not exist in the dataset (Li et al., 2024; Yang & Wang, 2025; Yuan et al., 2025). Recently, several concurrent works have investigated the potential of learning a world model by diffusion sampling. PolyGRAD (Rigter et al., 2024) and PGD (Jackson et al., 2024) introduce the diffusion model to model the transition function, and embed the policy for classifier-guided trajectory generation. In contrast, DWM (Ding et al., 2024) offers long-horizon predictions in a single forward pass, effectively reducing the compounding error and eliminating the need for recursive queries. In this paper, we follow the generation strategy in (Lu et al., 2023c; Liu et al., 2024; Wang et al., 2024; Lee et al., 2024), focus on adaptively synthesizing data for efficient offline-to-online RL fine-tuning.

## 6 CONCLUSION

In this paper, we propose AD2S, a diffusion-based data synthesizer for offline-to-online RL fine-tuning. With the data alignment and amplified guidance, AD2S can reuse high-novelty near on-policy data and enrich the data in high-rewarding regions. As a versatile solution, AD2S seamlessly integrates with prevalent offline-to-online frameworks, with no algorithmic modification. Our extensive

experiments on commonly studied benchmarks exhibit considerable performance improvements compared to other diffusion-based data synthesizers. We show that AD2S successfully generates high-quality data from ground-truth datasets, leading to a sample-efficient online fine-tuning.

## 7    ETHICS STATEMENT

This work adheres to the ICLR Code of Ethics. In this study, no human subjects or animal experimentation were involved. All datasets used were sourced in compliance with relevant usage guidelines, ensuring no violation of privacy. We have taken care to avoid any biases or discriminatory outcomes in our research process. No personally identifiable information was used, and no experiments were conducted that could raise privacy or security concerns. We are committed to maintaining transparency and integrity throughout the research process.

## 8    REPRODUCIBILITY STATEMENT

We have made every effort to ensure that the results presented in this paper are reproducible. The source code has been submitted in the supplementary material to facilitate replication and verification. We leave the detailed description of implementation details in Appendix A.

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

## A IMPLEMENTATION DETAILS

### A.1 TASK DESCRIPTION

**MuJoCo LocoMotion.** MuJoCo locomotion encompasses several standard locomotion tasks commonly utilized in RL research, such as Hopper, Halfcheetah, and Walker2d. In each task, the RL agent controls a robot to achieve forward movement. D4RL (Fu et al., 2020) benchmark provides four qualities of datasets for each task: random-v2, medium-v2, medium-replay-v2, medium-expert-v2.

**Maze2D** The Maze2D domain is a navigation task requiring a 2D agent to reach a fixed goal location. The tasks are designed to provide a simple test of the ability of offline RL algorithms to stitch together previously collected subtrajectories to find the shortest path to the evaluation goal. The variations of this environment can be initialized with different maze configurations and increasing levels of complexity Three maze layouts are provided: umaze, medium, and large. The task in the environment is for a 2-DoF ball that is force-actuated in the cartesian directions x and y, to reach a target goal in a closed maze.

**AntMaze Navigation.** Our tests on AntMaze navigation benchmark consist of 4 datasets, namely umaze-diverse-v2, medium-play-v2, medium-diverse-v2, and large-play-v2 from D4RL (Fu et al., 2020). The objective is for an ant to learn how to walk and navigate from the starting point to the destination in a maze environment, with only sparse rewards provided. This task poses a challenge for online RL algorithms to explore high-quality data effectively without access to offline datasets or additional domain knowledge.

**Adroit Manipulation.** Our empirical evaluation on Adroit manipulation contains 2 domains: pen, door. The RL agent is required to solve dexterous manipulation tasks, including rotating a pen in specific directions, opening a door, and moving a ball, respectively. The offline datasets are clone-v1 datasets in D4RL Fu et al. (2020) benchmark, which only contain a few successful non-Markovian human demonstrations. Therefore, it is pretty difficult for most offline RL approaches to acquire reasonable pre-training performances.

### A.2 IMPLEMENTATIONS AND HYPERPARAMETERS IN AD2S

Our Cal-QL implementation is based on previous work (Liu et al., 2024; Tarasov et al., 2023b), and primarily followed their recommended RL algorithm settings. The code can be found at `https://github.com/tinkoff-ai/CORL` and `https://github.com/liuxhym/EDIS`, which are released under an Apache license. The hyperparameters used in our AD2S's other module are detailed in the Table 4. Our method only introduces two MLP models on top of the diffusion model with corresponding sampling ratio to calculate condition, and we keep the same parameters across different tasks in each domain (e,g, Walker2d, Hopper, Halfcheetah in locomotion domain). On the contrary, EDIS has to train three independent energy models and tune three grad scales for each task. We believe that our method does not require an obvious computational budget and is easy to find the optimal sampling ratio in different tasks.

For all diffusion-based baselines, we use a 6-layer residual MLP as the denoising network. The residual denoising MLP not only provides high-fidelity data generation, but also enables a friendly computational cost in the online fine-tuning stage compared to other popular denoising networks such as U-net (Lee et al., 2024) or transformer (He et al., 2023a). During online fine-tuning, the diffusion synthesizer is retrained on offline and online samples for every $10,000$ environment steps. We also use $5,000$ steps at the start of online fine-tuning to warm up the online replay buffer in AD2S and PGR. For the diffusion sampling process, we follow previous works (Lu et al., 2023c; Wang et al., 2024; Liu et al., 2024), using the stochastic SDE sampler of Karras et al. (Karras et al., 2022) with the same hyperparameter used in EDIS (Liu et al., 2024).

**Computation Resources** We train AD2S integrated with the base algorithm on an NVIDIA GeForce RTX 3090 GPU and a 32-core CPU.

Table 4: Hyperparameters of AD2S for offline-to-online RL.

| Hyperparameter | Setting |
|---|---|
| Network Type (Denoising) | Residual MLP |
| Denoising Network Depth | 6 layers |
| Denoising Steps | 128 steps |
| Denoising Network Learning Rate | $3 \times 10^{-4}$ |
| Denoising Network Hidden Dimension | 1024 units |
| Denoising Network Batch Size | 256 |
| Denoising Network Activation Function | ReLU |
| Denoising Network Optimizer | Adam |
| CFG Scale | 2.0 |
| Condition Dropout Rate | 0.25 |
| Learning Rate Schedule (Denoising Network) | Cosine Annealing |
| Training Epochs (Denoising Network) | $50,000$ epochs |
| Training Interval Environment Step (Denoising Network) | Every $10,000$ steps |
| Replay Buffer Warm Up Step | $5,000$ steps |
| Density Ratio Network $w_\psi$ Hidden Dimension | 256 units |
| Density Ratio Network $w_\psi$ Activation Function | ReLU |
| Dynamics Prediction Network $g_\phi$ Hidden Dimension | 256 units |
| Dynamics Prediction Network $g_\phi$ Activation Function | Swish |
| $w_\psi$ & $g$ Learning Rate | $3 \times 10^{-4}$ |
| $w_\psi$ & $g$ Optimizer | Adam |
| Amplified Ratio $\alpha$ | 1.2 |
| Partial Noising Scale $\mu$ | 0.5 in Locomotoin & AntMaze 0.8 in Maze2D |
| Density-based Prioritized Sampling Ratio $p_{\text{DR}}$ | 0.5 in Locomotoin & AntMaze 0.8 in Maze2D |
| Curiosity-based Prioritized Sample Ratio $p_{\text{Curi}}$ | 0.5 in Locomotoin & AntMaze 0.8 in Maze2D |
| Advantage Weight for Density Ratio $\beta_{\text{DR}}$ | 10 |
| Advantage Weight for Curiosity $\beta_{\text{curi}}$ | 10 |

## B  ADDITIONAL EXPERIMENTS

### B.1  RESULTS ON OTHER ENVIRONMENTS

To evaluate AD2S in sparse-reward and complex environments, we conduct experiments on the AntMaze navigation and Adroit manipulation benchmarks. Empirical results (Table 5) demonstrate that AD2S consistently outperforms baseline methods in settings with sparse rewards and complex action spaces. However, we observe that all methods exhibit unstable online fine-tuning, which we attribute to the inherent challenges of the AntMaze and Adroit benchmarks.

### B.2  VERSATILITY OF AD2S

To show the versatility of AD2S, we integrate our method with IQL (Kostrikov et al., 2022) and WSRL (Zhou et al., 2024). Experimental results on the Walker2d environment (Table 6) demonstrate consistent performance improvements. These results validate AD2S's ability to enhance O2O RL across different baseline methods.

Table 5: D4RL normalized scores over five seeds on the antmaze and adroit tasks with the highest scores highlighted in **bold**. We also underline the AD2S's score when it is close to the best score in each task. We conduct experiments to demonstrate the performance of AD2S on sparse rewards and complex environments.

| Dataset | Cal-QL | EDIS | AD2S |
|---|---|---|---|
| antmaze-umaze-diverse-v2 | 93.4± 4.6 | 95.9± 2.8 | **96.8± 3.0** |
| antmaze-medium-play-v2 | 86.8± 1.6 | 93.9± 2.7 | **94.4± 5.2** |
| antmaze-medium-diverse-v2 | 81.4± 3.9 | **89.3± 4.8** | 85.0±10.0 |
| antmaze-large-play-v2 | 42.5± 5.2 | 66.1± 8.2 | **72.5±11.8** |
| antmaze-large-diverse-v2 | 42.3± 2.2 | 57.1± 2.8 | **64.0±11.4** |
| door-clone-v1 | -0.3± 0.1 | 55.8±25.7 | **78.0±17.6** |
| pen-clone-v1 | 10.7±10.2 | 81.7±14.9 | **93.9± 8.2** |

Table 6: D4RL normalized scores over five seeds on the walker2d task with 4 data qualities. We conduct experiments to show the versatility of AD2S by combining it with other backbone algorithms.

| Dataset | IQL | IQL + AD2S | WSRL | WSRL + AD2S |
|---|---|---|---|---|
| walker2d-random-v2 | 6.5± 0.7 | **12.1± 4.1** | 65.4±18.2 | **82.8±19.8** |
| walker2d-medium-v2 | 83.6± 2.0 | **98.2± 2.6** | 114.3± 4.5 | 112.3± 8.7 |
| walekr2d-medium-replay-v2 | 83.6± 2.1 | **93.6± 4.7** | 86.4±12.5 | **96.2±10.2** |
| walker2d-medium-expert-v2 | 108.9± 2.9 | **118.6± 1.3** | 118.8± 2.5 | **121.5± 1.6** |

Table 7: D4RL normalized scores over five seeds on the walker2d task with 4 data qualities. Here, we investigate the sensitivity of the distance alignment ratio $p_{\mathrm{DR}}$ in AD2S.

| Dataset | $p_{\mathrm{DR}} = 0.1$ | $p_{\mathrm{DR}} = 0.3$ | $p_{\mathrm{DR}} = 0.5$ | $p_{\mathrm{DR}} = 0.7$ |
|---|---|---|---|---|
| walker2d-random-v2 | 36.8±26.9 | 91.8± 2.5 | **99.3±11.6** | 62.4±23.6 |
| walker2d-medium-v2 | 86.1± 1.9 | 118.6± 2.8 | **119.4± 1.2** | 117.5± 1.7 |
| walekr2d-medium-replay-v2 | 120.3± 3.4 | 121.3± 3.7 | **121.0± 7.6** | 119.2± 6.7 |
| walker2d-medium-expert-v2 | 110.7± 1.4 | 123.6± 1.7 | **129.1± 4.6** | 119.8± 3.0 |

## C  ADDITIONAL ABLATION STUDY

### C.1  SENSITIVITY ANALYSIS.

**Distance-based alignment ratio.**  We conduct experiments on the Walker2d task with 4 dataset qualities to perform a sensitivity analysis on $p_{\mathrm{DR}}$ in AD2S. We choose $p_{\mathrm{DR}}$ from $[0.1, 0.3, 0.5, 0.7]$ and the results are presented in Table 7. As demonstrated in our results, simple grid search on $p_{\mathrm{DR}}$ is sufficient for tuning AD2S. The constrained alignment ratio narrows the range of reusable data (e.g., $p_{\mathrm{DR}} = 1$), thereby compromising the diversity of the synthesized distribution.

**Curiosity prioritization ratio.**  We also investigate the choice of $p_{\mathrm{Curi}}$ for walker2d task with 4 data qualities in Table 8 and choose 4 levels $p_{\mathrm{Curi}}$ from $[0.1, 0.3, 0.5, 0.7]$. The results demonstrate that the proposed AD2S does not require heavy hyperparameter tuning, and performs well reproducibility.

**Amplified scale.**  We provide the sensitivity analysis of $\alpha$ in AD2S on walker2d task with 4 dataset qualities and 5 levels $\alpha$ from $[0.8, 1.0, 1.2, 1.5, 2.0]$. Table 9 reveals that AD2S struggles to synthesize samples whose advantage-weighted data distributions are distant from the ground-truth data, especially on a low-quality dataset. Furthermore, our analysis demonstrates that while moderate conditioning amplification improves performance on medium- and high-quality datasets, a more conservative conditional guidance yields better results for low-quality datasets.

Table 8: D4RL normalized scores over five seeds on the walker2d task with 4 data qualities. Here, we investigate the sensitivity of the curiosity alignment ratio $p_{\text{Curi}}$ in AD2S.

| Dataset | $p_{\text{Curi}} = 0.1$ | $p_{\text{Curi}} = 0.3$ | $p_{\text{Curi}} = 0.5$ | $p_{\text{Curi}} = 0.7$ |
|---|---|---|---|---|
| walker2d-random-v2 | 78.9± 7.6 | 77.4± 1.6 | **99.3±11.6** | 81.6± 9.8 |
| walker2d-medium-v2 | 117.7± 4.9 | 119.4± 3.0 | **119.4± 1.2** | 117.8± 4.9 |
| walekr2d-medium-replay-v2 | 119.7± 2.5 | 119.2± 5.4 | **121.0± 7.6** | 114.8± 2.5 |
| walker2d-medium-expert-v2 | 124.2± 1.2 | 124.7± 4.1 | **129.1± 4.6** | 123.4± 1.8 |

Table 9: D4RL normalized scores over five seeds on the walker2d task with 4 data qualities. We conduct experiments to investigate the sensitivity of amplified scale $\alpha$ in AD2S.

| Dataset | $\alpha = 1.0$ | $\alpha = 1.2$ | $\alpha = 1.5$ | $\alpha = 2.0$ |
|---|---|---|---|---|
| walker2d-random-v2 | 89.1±15.6 | **99.3±11.6** | 90.8± 5.2 | 74.4± 2.5 |
| walker2d-medium-v2 | 110.8± 1.2 | **119.4± 1.2** | 108.5± 5.2 | 104.9± 3.7 |
| walekr2d-medium-replay-v2 | 117.5± 5.7 | **121.0± 7.6** | 119.4± 4.6 | 117.1± 6.1 |
| walker2d-medium-expert-v2 | 120.8± 2.5 | **129.1± 4.6** | 120.5± 3.6 | 120.6± 4.0 |

Table 10: D4RL normalized scores over five seeds on the walker2d task with 4 data qualities. We conduct experiments to investigate the sensitivity of the advantage temperature $\beta$ in AD2S.

| Dataset | $\beta = 1$ | $\beta = 10$ | $\beta = 100$ |
|---|---|---|---|
| walker2d-random-v2 | 85.0±11.7 | **99.3±11.6** | 16.4± 4.9 |
| walker2d-medium-v2 | 119.2± 5.8 | **119.4± 1.2** | 85.1± 1.8 |
| walekr2d-medium-replay-v2 | 119.7± 8.8 | **121.0± 7.6** | 72.5±19.8 |
| walker2d-medium-expert-v2 | 127.9± 1.9 | **129.1± 4.6** | 113.5± 6.9 |

Table 11: D4RL normalized scores over five seeds on the walker2d-medium task. We conduct experiments to validate the efficacy of our proposed advantage-weighted alignment.

| Dataset | AD2S | DR w/o Adv. | Curiosity w/o Adv. |
|---|---|---|---|
| walker2d-medium-v2 | **119.4± 1.2** | 103.1± 8.5 | 106.7± 2.3 |

**Advantage temperature.** For the advantage temperature $\beta$, we conduct an ablation study across values on the Walker2d task and present the results in Table 10. In AD2S, the role of $A(\cdot, \cdot)$ is to find out transitions in the aligned data that possess high policy improvement potential. Empirical findings indicate $\beta = 10$ delivers optimal performance, and there is little difference between $\beta = 1$ and $\beta = 10$ on *medium*, *medium-expert*, and *medium-replay*, while $\beta = 100$ causes obvious performance degradation. Therefore, our method only needs to ensure that the scaled advantage does not dominate either the density ratio or the curiosity metric. In our experimental settings, we maintain a consistent temperature for all tasks within the same benchmark domain (i.e., the locomotion tasks: Walker2d, Hopper, and HalfCheetah).

## C.2 ADVANTAGE-WEIGHTED ALIGNMENT

To illustrate the reason for using the relative advantage metric in both steps, we conduct experiments on the walker2d-medium task with two variants of AD2S: density ratio without advantage (**DR w/o Adv.**) and curiosity without advantage (**Curiosity w/o Adv.**). Results in Table 11 validate the effectiveness of our method. Both the density ratio and the curiosity measurement lack regard for the value of a sample held in the RL environment, and the advantage estimation reflects the potential improvement that the current sample can bring to the policy. Thus, we incorporated the advantage metric in both steps to improve online fine-tuning efficiency.

Table 12: Results on v-d4rl (Lu et al., 2023b) over five seeds on the Walker Walk task with 3 data qualities. We highlight the highest scores in **bold**.

| Dataset | DrQ-BC | SynthER | PGR | AD2S |
|---|---|---|---|---|
| random | 459.7±29.0 | 484.2±52.6 | 470.8±47.0 | **510.1±28.5** |
| medium | 490.6±18.5 | 494.1±50.9 | 530.9±23.1 | **527.0±38.7** |
| medium-replay | 488.7±32.7 | 490.2±56.7 | 503.0±5.3 | **508.7±14.1** |

### C.3 EXPERIMENTS ON VISUAL ENVIRONMENT

We conduct experiments on pixel-based environments using the v-d4rl benchmark (Lu et al., 2023b), following the data generation paradigm proposed by Lu et al. (2023c). Firstly, we pretrain the DrQ-BC for 1M steps on the offline dataset. Then we freeze the image encoder, generate latent observations using diffusion models, and fine-tune the policy and Q-network in the online environment for 200k steps. We use the same architecture and hyperparameters as used in the D4RL locomotion benchmark, other than changing the partial noising scale $\mu$ to 0.1, amplified ratio $\alpha$ to 2.0, and CFG scale to 1.0. Table 12 shows the results on the Walker Walker task with 3 datasets. In future work, we will explore even more complex environments to refine our method and validate its generalization.

### C.4 SYNTHETIC DATA ANALYSIS

To verify the validity of the synthetic samples generated by AD2S, we use the ground-truth simulator to measure the dynamics distance (i.e., MSE error) between AD2S or other diffusion-based baselines with the real next state. Empirical results are presented in Figures 5. These measurements provide insight into the efficacy of our proposed method.

While AD2S incurs a higher dynamic distance due to its curiosity-driven data prioritization, it synthesizes data further than all baseline methods in the distance metric. This capability helps mitigate the inherent pessimism in offline-pretrained Q-functions and improves online exploration. This demonstrates that AD2S is better at pushing synthetic data towards regions with higher novelty.

## D ADDITIONAL EXPERIMENTS

### D.1 ONLINE LEARNING CURVES

We plot the online learning curves for three MuJoCo control tasks in Figure 6 to characterize the agent's behavior and reveal sample-efficiency trends during online fine-tuning. The results show that AD2S improves the sample-efficiency of online fune-tuning compared to other diffusion-based baselines.

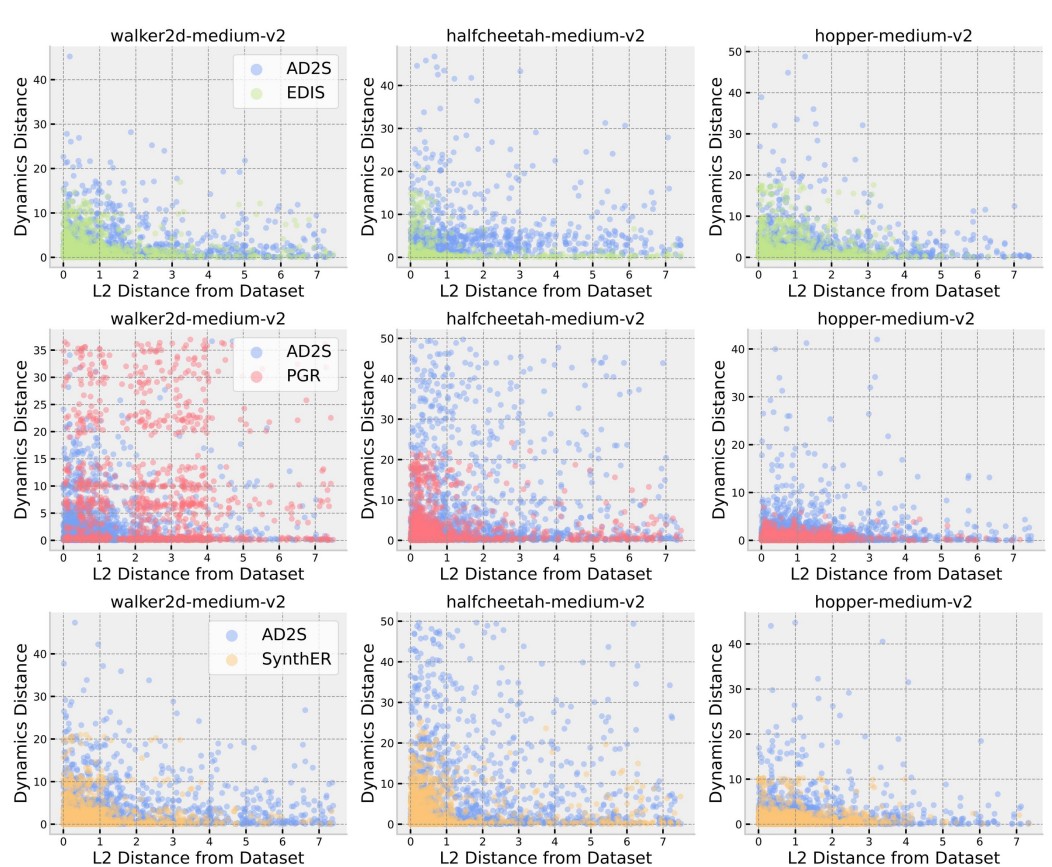

Figure 5: We plot L2 distance from online collected data, and dynamic distance under AD2S or diffusion-based baselines. **Top**: EDIS, **Middle**: PGR, and **Bottom**: SynthER. AD2S can adaptively generate higher novelty data than the existing SOTA method This indicates that our method can adaptively generate higher novelty data than the existing SOTA method.

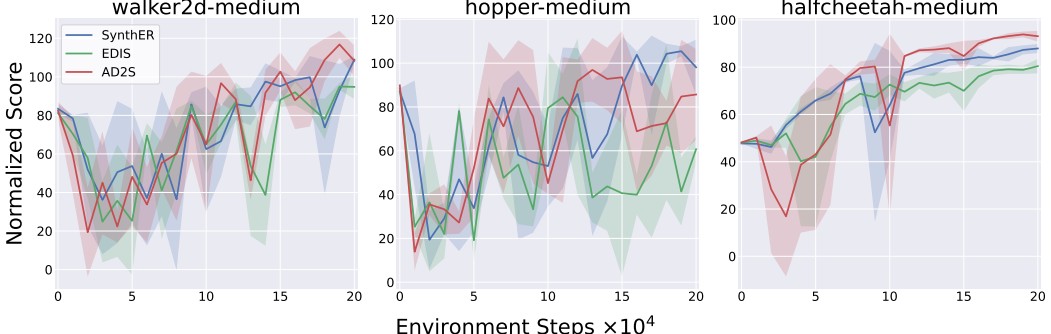

Figure 6: Online learning curves on three Mujoco control tasks (walker2d-medium, hopper-medium, and halfcheetah-medium).

