# OpenReview forum: "Enhancing Offline-to-Online Reinforcement Learning by Adaptive Experience Aligned Diffusion Sampling"
_ICLR.cc/2026/Conference — Submitted to ICLR 2026_

### Official Review · Reviewer_pLsX · 2025-10-15

**Soundness:** 3
**Presentation:** 3
**Contribution:** 2
**Rating:** 4
**Confidence:** 4

**Summary:**

This paper introduces Adaptive Data Aligned Diffusion Sampling (AD2S), a novel method for enhancing offline-to-online reinforcement learning. The core idea is to use a diffusion model to generate synthetic data that accelerates the fine-tuning of an agent pre-trained on an offline dataset. AD2S is designed to be a plug-in approach that can be combined with existing offline-to-online RL algorithms. The paper demonstrates its effectiveness by integrating it with Cal-QL, a state-of-the-art method in this domain.

**Strengths:**

The paper’s primary strength is its direct and intelligent approach to solving the fundamental challenge of offline-to-online (O2O) fine-tuning. The authors correctly identify the tension between the conservative nature of offline-trained agents and the need for efficient online exploration. Instead of naively replaying offline data, the method provides a sophisticated mechanism to generate targeted, "near on-policy" synthetic data. This is a more reasonable and effective strategy for bridging the significant distribution gap between the offline and online settings. The three-stage pipeline (alignment, prioritization, and regeneration) is a logical way to ensure the generated data is actively steered towards high-reward and high-novelty regions.

**Weaknesses:**

1. While the proposed pipeline is effective, its overall contribution feels more incremental than foundational. The paper essentially combines several well-established techniques from different areas of reinforcement learning into a single framework. Since the use of diffusion models for data generation in O2O scenarios, curiosity-driven learning, and importance weighting are all well-developed domains, this paper comes across as a strong piece of engineering work but lacks significant technical originality.
2. The AD2S pipeline is a complex system, which raises my practical concerns about its usability and efficiency.
3. I hope the authors can add some real-world validation or extend the O2O to domains like large language models.

**Questions:**

See weaknesses.

---

> ### Author Response · Authors · 2025-11-23
>
> Thank you for your time and feedback on AD2S. We address each of your questions below.
>
> **W1:** This paper comes across as a strong piece of engineering work but lacks significant technical originality.
>
> **A:** For the diffusion-based methods that focus on O2O RL, EDIS [1] introduces several energy models to guide the diffusion model for generating online distribution data, and CFDG [2] applies a re-weighting trick to balance the offline and online synthetic data during online fine-tuning. Compared to previous methods, we not only require the diffusion model to re-generate synthetic data from historical near on-policy transitions but also push these data towards high-rewarding and underexplored regions. This helps stabilize the early online Q-learning process and encourages the agent to explore the environment and calibrate the Q network.
>
> **W2:** The AD2S pipeline is a complex system, which raises my practical concerns about its usability and efficiency.
>
> **A:** Our method introduces only two MLP models on top of the diffusion model, along with a corresponding sampling ratio to calculate the condition. We maintain the same parameters across different tasks in each domain (e.g., Walker2d, Hopper, Halfcheetah in the locomotion domain). During online fine-tuning, the update of two MLP models adds only 0.04 seconds per step, and the Cal-QL algorithm operates at 0.3 seconds per step. This represents a marginal increase in computational cost, demonstrating that the additional models impose a negligible computational burden.
>
> We also conduct experiments to test an imperfect data generator by introducing early-stopped diffusion models (AD2S-20% and AD2S-50% with 20% and 50% of default training steps, respectively). The results shown below demonstrate that our method still improves online fine-tuning when the diffusion models are not trained well, indicating the robustness and usability of AD2S.
>
> |          | Cal-QL | AD2S | AD2S-20% | AD2S-50% |
> | -------- | --- | --- | --- | --- |
> | walker2d-random | 15.1± 3.5 | **119.4± 1.2** | 89.0±7.0 | 88.1±7.8 |
> | walker2d-medium | 84.2± 0.3 | **99.3±11.6** | 95.8±9.6 | 96.1±9.6 |
>
>
> **W3:** I hope the authors can add some real-world validation or extend the O2O to domains like large language models.
>
> **A:** For the LLM domains, our method could utilize the pre-trained dataset and specific in-domain data to train the diffusion model for text generation. Based on the synthetic data, the LLM could be fine-tuned on the domain-specific data, which is theoretically feasible.
> Due to the limited time and experimental resources, we are trying to conduct experiments on high-dimensional control tasks such as the Adroit Dexterous benchmark, which closely resemble real-world scenarios. We will try our best to provide the experimental results during the subsequent discussion to the best of our ability.
>
> [1] Energy-Guided Diffusion Sampling for Offline-to-Online Reinforcement Learning
>
> [2] Offline-to-Online Reinforcement Learning with Classifier-Free Diffusion Generation

---

### Official Review · Reviewer_BJwC · 2025-10-26

**Soundness:** 3
**Presentation:** 3
**Contribution:** 3
**Rating:** 6
**Confidence:** 2

**Summary:**

This paper proposes AD2S (Adaptive Data-Aligned Diffusion Sampling) to improve offline-to-online reinforcement learning by generating high-quality, task-aligned synthetic data. AD2S adaptively aligns offline data with online samples, prioritizes informative experiences via curiosity, and uses a guided diffusion model to regenerate data in promising regions. Experiments show consistent performance gains over existing baselines.

**Strengths:**

1. Originality:
Combines data alignment, curiosity-based prioritization, and guided diffusion sampling into a unified framework for offline-to-online RL.

2. Quality:
Methodologically sound with coherent design addressing distribution shift and exploration. Experiments on diverse D4RL tasks show consistent and meaningful gains over baselines.

3. Clarity:
Well-organized and clearly written.

4. Significance:
Provides a practical, general, and plug-in approach that improves sample efficiency and stability in O2O RL, with strong potential for broader real-world impact.

**Weaknesses:**

1. Lack of theoretical grounding:
The method is intuitively motivated but lacks formal theoretical analysis of its effect on sample efficiency.


2. Dependence on model quality:
Performance may degrade with imperfect dynamics or diffusion models; robustness analysis is missing.

3. Computational cost:
Diffusion sampling introduces extra overhead, but runtime comparisons with lighter baselines are not provided.

**Questions:**

1. How sensitive is performance to imperfect diffusion models?

2. What is the computational overhead compared to simpler baselines?

---

> ### Author Response · Authors · 2025-11-23
>
> Thank you for your time and feedback on AD2S. We address each of your questions below.
>
> **W1:** The method is intuitively motivated but lacks formal theoretical analysis of its effect on sample efficiency.
>
> **A:** We thank the reviewer for the insightful comment regarding the theoretical grounding of our method. We agree that providing a deeper theoretical understanding is crucial. Our method was indeed inspired by Lee et al. [1], where Liu et al. [2] give a theoretical lower bound on the performance improvement gap under the advantage-weighted data alignment. This lower bound guarantees the training stability of the agent, which is crucial for online fine-tuning in O2O RL. To further accelerate the online exploration, we require the diffusion model to generate high-rewarding, underexplore data. The gradients of the squared objective in Eq. 9 with the prioritized sampling are equivalent to the cubic power objective $\frac{1}{3n} \sum_{i=1}^{n} |s' - \hat{s}'|^3$ under a random sample [3]. This analysis highlights that curiosity-driven data generation encourages the agent to explore the online dynamics, thereby enabling a sample-efficient online fine-tuning.
>
>
> **W2:** Dependence on model quality, robustness analysis is missing. How sensitive is performance to imperfect diffusion models?
>
> **A:** We conduct experiments of introducing early stopped diffusion models during online fine-tuning (AD2S-20% and AD2S-50% with 20% and 50% of default training steps, respectively) to test the sensitivity of AD2S to imperfect diffusion models. The results are presented below. We find that our method still improves performance when the diffusion models are not trained well, indicating the robustness and usability of AD2S. These results indicate that our method can improve the lower limit of the O2O performance, and the overall training process is controllable with an appropriate training process of diffusion models.
>
> |          | Cal-QL | AD2S | AD2S-20% | AD2S-50% |
> | -------- | --- | --- | --- | --- |
> | walker2d-random | 15.1± 3.5 | **119.4± 1.2** | 89.0±7.0 | 88.1±7.8 |
> | walker2d-medium | 84.2± 0.3 | **99.3±11.6** | 95.8±9.6 | 96.1±9.6 |
>
> **W3:** Computational cost: Diffusion sampling introduces extra overhead, but runtime comparisons with lighter baselines are not provided. What is the computational overhead compared to simpler baselines?
>
> **A:** Our method introduces only two MLP models on top of the diffusion model, along with a corresponding sampling ratio to calculate the condition. We maintain the same parameters across different tasks in each domain (e.g., Walker2d, Hopper, Halfcheetah in the locomotion domain). During online fine-tuning, the update of two MLP models adds only 0.04 seconds per step, and the Cal-QL algorithm operates at 0.3 seconds per step. This represents a marginal increase in computational cost, demonstrating that the additional models impose a negligible computational burden.
>
> [1] Offline-to-Online Reinforcement Learning via Balanced Replay and Pessimistic Q-Ensemble
>
> [2] Active Advantage-Aligned Online Reinforcement Learning with Offline Data
>
> [3] Understanding and Mitigating the Limitations of Prioritized Experience Replay

---

### Official Review · Reviewer_RqZB · 2025-10-27

**Soundness:** 2
**Presentation:** 2
**Contribution:** 2
**Rating:** 2
**Confidence:** 4

**Summary:**

This paper introduces AD2S (Adaptive Data Aligned Diffusion Sampling), a plug-and-play data generation framework for offline-to-online reinforcement learning (O2O RL). A2DS aligns near on-policy transitions via density ratios, prioritizes high-curiosity/high-reward samples, and regenerates synthetic data using a diffusion model with amplified guidance. Evaluated on D4RL MuJoCo and Maze2D benchmarks, AD2S consistently improves upon strong baselines like Cal-QL, especially on low-quality datasets.

**Strengths:**

- This paper proposes AD2S, a hybrid-driven data generation method combining advantage and curiosity signals to improve policy performance in offline-to-online RL.
- AD2S demonstrates strong performance on MuJoCo tasks, achieving state-of-the-art results across most tasks.
- The ablation studies are comprehensive, effectively validating the contribution of each module and the choice of hyperparameters.

**Weaknesses:**

- The writing of this paper needs improvement. Key notations and formal definitions are missing (e.g., $\mathcal{D}^\text{re}$, $\mathcal{D}^\text{aligned}$, and the detailed loss function in Equation 7), making it difficult to follow the method section until after reading the framework summary. This hinders understanding of AD2S’s algorithmic flow and implementation details.
- The proposed statistical relative advantage $A(s, a) = (r - r_\text{mean}) / r_\text{std}$ does not reflect true advantage, as $r_\text{mean}$ and $r_\text{std}$ are constants, resulting in a policy-agnostic normalized reward rather than a policy-dependent advantage. In sparse-reward settings (e.g., AntMaze), this estimator may collapse to near-zero for most transitions, rendering advantage weighting ineffective.
- The evaluation is limited to MuJoCo tasks. Performance on sparse-reward or pixel-based domains remains unverified.
- The paper lacks an analysis of additional computational costs introduced by each component of the algorithm.
- The authors should also evaluate sample efficiency (e.g., online training return curves) and whether performance drops occur during the offline-to-online transition. These critical metrics are missing from the experiments.

**Questions:**

- How does AD2S perform in sparse-reward environments (e.g., AntMaze), where the relative advantage estimator may collapse?
- Can the framework be extended to pixel-based or visual RL, where reward estimation and density ratio modeling are significantly more challenging?
- What is the computational cost per online step when using AD2S?
- How does AD2S perform in terms of sample efficiency? Can it mitigate performance drops when policies switch from offline to online settings?

---

> ### Author Response · Authors · 2025-11-23
>
> Thank you for your time and feedback on AD2S. We address each of your questions below.
>
> **W1:** The writing of this paper needs improvement.
>
> **A:** Thanks for your thoughtful suggestion. We have already added key notations and formal definitions of components in AD2S in section 3.1 in the revised paper.
>
> **W2:** In sparse-reward settings (e.g., AntMaze), this single-step estimator may collapse to near-zero for most transitions, rendering advantage weighting ineffective.
>
> **A:**  We consider our single-step estimator as a complementary term during the two-step data alignment process. These immediate reward signals are formed as a bonus that encourages the agent to explore transitions induced by these high relative-reward transitions. In the sparse reward task, when the estimator collapses to near-zero, our method remains driven by data alignment and the curiosity term. To validate our single-step estimator on the sparse reward task, we compare against an AD2S variant using the learned Q-network: $A(s,a) = Q(s,a) - \mathbb{E} [Q(s, \tilde{a})]$ where  are random actions. Experimental results on the AntMaze tasks (shown below) demonstrate that our method yields superior performance compared to this Q-based approximation. This may be because the learned advantage exhibits instability during the initial stage of online fine-tuning, thus degrading the performance.
>
> |      Task          |   AntMaze-medium-play  |  AntMaze-medium-diverse  |  AntMaze-large-play  |
> | -------------- | --- | --- | --- |
> | AD2S           | 94.4±5.2 | **85.0±10.0** | **72.5±11.8** |
> | AD2S (variant) | **96.7±4.7** | 36.7±45.0 | 40.0±21.6 |
>
> **W3:** The evaluation is limited to MuJoCo tasks. Performance on sparse-reward or pixel-based domains remains unverified.
>
> **A:** To validate the efficacy of our approach in other domains, we present experimental results on the sparse-reward task (e.g., AntMaze) and pixel-based environment (V-D4RL) below, which have already been presented in Appendix B.
>
> | Sparse reward             | Cal-QL    | EDIS         | AD2S          |
> | ------------------------- | --------- | ------------ | ------------- |
> | antmaze-medium-play-v2    | 86.8±1.6  | 93.9±2.7     | **94.4±5.2**  |
> | antmaze-medium-diverse-v2 | 81.4±3.9  | **89.3±4.8** | 85.0±10.0     |
> | antmaze-large-play-v2     | 42.5±5.2  | 66.1±8.2     | **72.5±11.8** |
> | antmaze-large-diverse-v2  | 42.3±2.2  | 57.1±2.8     | **64.0±11.4** |
> | door-clone-v1             | -0.3±0.1  | 55.8±25.7    | **78.0±17.6** |
> | pen-clone-v1              | 10.7±10.2 | 81.7±14.9    | **93.9±8.2**  |
>
> | V-D4RL                    | DrQ-BC     | SynthER    | PGR        | AD2S           |
> | ------------------------- | ---------- | ---------- | ---------- | -------------- |
> | walker-walk random        | 459.7±29.0 | 484.2±52.6 | 470.8±47.0 | **510.1±28.5** |
> | walker-walk medium        | 490.6±18.5 | 494.1±50.9 | 530.9±23.1 | **527.0±38.7** |
> | walker-walk medium-expert | 488.7±32.7 | 490.2±56.7 | 503.0±5.3  | **508.7±14.1** |
>
> **W4:** The paper lacks an analysis of additional computational costs introduced by each component of the algorithm.
>
> **A:** Our method introduces only two MLP models on top of the diffusion model, along with corresponding sampling ratios to calculate the condition. We maintain the same parameters across different tasks in each domain (e.g., Walker2d, Hopper, Halfcheetah in the locomotion domain). During online fine-tuning, the update of two MLP models add only 0.04 seconds per step, and the Cal-QL algorithm operates at 0.3 seconds per step. This represents a marginal increase in computational cost, demonstrating that the additional models impose a negligible computational burden.
>
> **W5:** The authors should also evaluate sample efficiency (e.g., online training return curves) and whether performance drops occur during the offline-to-online transition.
>
> **A:** We have plotted the online learning curves on MuJoco control tasks in Appendix D.1 in the revised paper to characterize the agent's behavior and reveal sample-efficiency trends during online fine-tuning. The results show that AD2S improves the sample-efficiency of online fune-tuning compared to other diffusion-based baselines.

---

> > ### Comment · Reviewer_RqZB · 2025-11-27
> >
> > Thank you for your detailed response and for supplementing the experiments on sparse reward and pixel-based tasks. While I appreciate the additional effort, I still have several concerns:
> > 1. **Advantage Definition Issue**: Using reward $r$ as advantage makes it policy-independent, contradicting the claim (line 202) that it guides policy improvement, where advantage should be defined w.r.t. the current policy $\pi$.
> > 2. **Invalid Advantage Estimation**: Estimating $V^\pi(s)$ as $\mathbb{E}_{\tilde{a}\sim\text{Uniform}(\mathcal{A})}[Q(s,\tilde{a})]$ is incorrect. The baseline $V^\pi(s)$ requires actions sampled from $\pi$, not uniformly at random.
> > 3. **Limited Online Gains and Large Performance Drop**: Figure 6 shows that AD2S achieves only marginal improvements in online sample efficiency, while suffering the largest performance drop during the offline-to-online transition. Given the method’s considerable design complexity, this level of gain appears insufficient to justify the added overhead.
> > 4. **Inconsistent Results (Figure 6 vs. Table 1)**: In Figure 6, SynthER appears best on hopper-medium, yet Table 1 shows AD2S outperforming SynthER on this task.

---

> > > ### Author Response · Authors · 2025-12-01
> > >
> > > Thank you for your time and feedback on our paper. We address each of your concerns below.
> > >
> > > **Q1:** Advantage definition issue.
> > >
> > > **A:** We introduce advantage estimation during data alignment to quantify each sample's quality relative to other data in the entire data buffer. We then generate high-reward synthetic data to improve the quality of the online replay buffer. Most advantage estimations are policy dependent (e.g., $A(s,a) = Q(s,a) - \mathbb{E}_{\hat{a}\sim\pi}[Q(s, \hat{a})]$). However, the Q network during the early online fine-tuning process is unstable, leading to biased and inaccurate weighting. Moreover, the policy learned from the inaccurate will output sub-optimal actions, further disturbing the data sampling process. On the contrary, the policy-independent advantage in Eq. (8) provides a stable, computationally efficient way to identify high-reward samples across the entire data buffer. Improving the sampling ratio for single-step high-reward data will encourage the agent to revise its value estimates for these transitions, enabling a more efficient online exploration.
> > >
> > > **Q2:** Invalid advantage estimation.
> > >
> > > **A:** To further demonstrate our method's effectiveness and address your concern, we use advantage estimation based on the current policy and the Q network (named **AD2S with $A^\pi$**) and conduct experiments on the AntMaze task. The results in the table below show that our proposed advantage estimation in Eq. (8) improves the performance on complex tasks while remaining competitive on medium-difficulty tasks. The policy-dependent advantage estimation outputs an unreliable long-term estimate during early fine-tuning, thereby failing to weight the aligned data appropriately. Meanwhile, our proposed policy-independent advantage will output the same weight for states with near-zero reward, and the data alignment process will be dominated by the density ratio or curiosity, thereby continuing to generate high-quality data into the online replay buffer.
> > >
> > > | Sparse reward             | AD2S          | AD2S with $A^\pi$ |
> > > | ------------------------- | ------------- | ------------- |
> > > | antmaze-medium-play-v2    | **94.4±5.2**  | 93.3±5.0 |
> > > | antmaze-medium-diverse-v2 | 85.0±10.0     | **88.0±5.9** |
> > > | antmaze-large-play-v2     | **72.5±11.8** | 30.7±17.0 |
> > > | antmaze-large-diverse-v2  | **64.0±11.4** | 40.7±10.4 |
> > >
> > > **Q3:** Limited online gains and large performance drop.
> > >
> > > **A:** Our motivation is to generate synthetic data that is beneficial for online fine-tuning and obtain the best test performance. The performance drop during the early online stage is caused by the inaccurate offline pretrained Q network, which is a common issue in the O2O RL transition. However, our method is able to calibrate the offline pretrained model and obtain the highest best-score more efficiently. Compared to EDIS, our method gains nearly 9% improvement on Mujoco tasks and 11% improvement on AntMaze large tasks. Moreover, compared to SynthER, our method adds only two MLP models, whereas PGR adds one, and EDIS adds three. Based on the empirical results, we believe the added overhead is reasonable.
> > >
> > > **Q4:** Inconsistent results in hopper-medium task (Figure 6 vs. Table 1).
> > >
> > > **A:** Because we did not save the eval results for every 10000 steps when presenting Table 1, we reran the experiments to plot Figure 6. Due to time constraints, these results were run on 3 random seeds. This introduces randomness, leading to biased results. On the contrary, the results in Table 1 are run on 5 random seeds, which are more convincing. We will conduct experiments with more random seeds to provide a more convincing online learning curve.

---

### Official Review · Reviewer_wo2Y · 2025-10-31

**Soundness:** 3
**Presentation:** 3
**Contribution:** 3
**Rating:** 4
**Confidence:** 3

**Summary:**

This paper proposes Adaptive Data Aligned Diffusion Sampling (AD2S), a diffusion-based data augmentation method for offline-to-online (O2O) RL. The proposed method aims to accelerate online fine-tuning by generating synthetic transitions that are both high-rewarding and near on-policy. To achieve this, they design a three-step process: (1) distance-based experience alignment using a density ratio to select transitions that are close to on-policy distribution, (2) curiosity-driven data prioritization to identify novel and informative transitions, and (3) diffusion-based regeneration of synthetic data with amplified condition guidance to push the synthetic data toward high-rewarding regions. Experimental results, implemented on top of Cal-QL, show that the proposed method improves the sample efficiency of online fine-tuning.

**Strengths:**

S1. (Adaptive data reuse)
The proposed alignment process identifies near on-policy and high-rewarding transitions by combining density-ratio estimation with standardized advantage weighting. This alignment mechanism is conceptually sound for stabilizing Q-learning during online fine-tuning.

S2. (Validation of synthetic data quality)
By employing curiosity-based prioritization and amplified guidance in the diffusion process, the proposed processes can synthesize data in under-explored, high-reward regions. Visual analyses using t-SNE embeddings and oracle rewards (Figure 4) indicate that the proposed method generates data closer to high-reward, high-novelty regions compared to the baselines, also synthesizing data that is distinct from both the offline dataset and online replay buffer

S3. (plug-in compatibility)
The proposed method can serve as a plug-in generator that augments existing O2O RL frameworks without modifying the backbone algorithm. Experimental results (Tables 1, 6) show that the method consistently improves performance across existing O2O RL baselines.

**Weaknesses:**

W1. (Evaluation on limited domains)
Although results are reported on MuJoCo and Maze2D, several challenging sparse or semi-sparse reward MDPs (e.g., antmaze-umaze-v2, pen-human-v1, door-human-v1, and OGBench[1]) are not reported. Evaluations on these MDPs would more convincingly establish the generality of the proposed method. In addition, providing learning-curve plots would help characterize the behavior of the agent and reveal sample-efficiency trends during online fine-tuning.

W2. (Ambiguous effectiveness of the advantage estimator)
The proposed method replaces learned $Q-V$ advantages with a standardized reward during fine-tuning to improve stability. This standardized reward (eq. 8) is defined by a single-step reward and may not correlate with long-horizon performance in sparse or semi-sparse MDPs. For example, in binary-reward MDPs such as Antmaze, most transitions receive zero reward; it is unclear whether the standardized reward signal can reliably prioritize transitions that lead to goal-reaching behaviors. Broader analyses on sparse reward MDPs are needed to validate this advantage estimation design choice and to quantify its impact on the quality of synthesized data.

W3. (Missing algorithmic details)
While the three-step process is clearly presented at a high level, several operational specifics are insufficiently described. In particular, the roles and usage of $P_{DR}$ and $P_{Curi}$ in Table 4 are not fully specified.


**Minor**

- typo at line 124: $\nabla_x\log p(x;\sigma k)$ -> $\nabla_x\log p(x;\sigma(k))$

- ambiguous notation at line 258: $\mu(0<\mu...)$

[1] Park, Seohong, et al. "Ogbench: Benchmarking offline goal-conditioned rl." arXiv preprint arXiv:2410.20092 (2024).

**Questions:**

Could you provide results or analyses on the weaknesses? In particular, it would be valuable to include experiments or discussions on sparse or semi-sparse reward domains.

---

> ### Author Response · Authors · 2025-11-23
>
> Thank you for your time and feedback on AD2S. We address each of your questions below.
>
> **W1:** Evaluation on limited domains. (sparse reward, semi-sparse reward; learning-curve plots during online learning)
>
> **A:** We present experimental results (we have already given in Appendix B.1) on sparse-reward tasks below, benchmarking against both EDIS and baseline methods.
>
> | Dataset                   | Cal-QL    | EDIS         | AD2S          |
> | ------------------------- | --------- | ------------ | ------------- |
> | antmaze-medium-play-v2    | 86.8±1.6  | 93.9±2.7     | **94.4±5.2**  |
> | antmaze-medium-diverse-v2 | 81.4±3.9  | **89.3±4.8** | 85.0±10.0     |
> | antmaze-large-play-v2     | 42.5±5.2  | 66.1±8.2     | **72.5±11.8** |
> | antmaze-large-diverse-v2  | 42.3±2.2  | 57.1±2.8     | **64.0±11.4** |
> | door-clone-v1             | -0.3±0.1  | 55.8±25.7    | **78.0±17.6** |
> | pen-clone-v1              | 10.7±10.2 | 81.7±14.9    | **93.9±8.2**  |
>
> We also plot the online learning curves on Mujoco control tasks in Appendix D.1 in the revised paper to characterize the agent's behavior and reveal sample-efficiency trends during online fine-tuning. The results show that AD2S improves the sample-efficiency of online fune-tuning compared to other diffusion-based baselines.
>
> **W2:** Ambiguous effectiveness of the advantage estimator. Broader analyses on sparse reward MDPs are needed to validate the proposed advantage estimation.
>
> **A:** We consider our advantage estimator as a complementary term during the two-step data alignment process. These immediate reward signals are formed as a bonus that encourages the agent to explore transitions induced by these high relative-reward transitions. In the sparse reward task, when the estimator collapses to near-zero, our method remains driven by data alignment and the curiosity term. To validate our advantage estimator on the sparse reward task, we compare against an AD2S variant using the learned Q-network: $A(s,a) = Q(s,a) - \mathbb{E} [Q(s, \tilde{a})]$ where $\tilde{a}$ are random actions. Experimental results on three AntMaze tasks (shown below) demonstrate that our method yields superior performance and training stability compared to this Q-based approximation. This may be because the learned advantage exhibits instability during the initial stage of online fine-tuning, which in turn degrades the performance.
>
> |      Task          |   AntMaze-medium-play  |  AntMaze-medium-diverse  |  AntMaze-large-play  |
> | -------------- | --- | --- | --- |
> | AD2S           | 94.4±5.2 | **85.0±10.0** | **72.5±11.8** |
> | AD2S (variant) | **96.7±4.7** | 36.7±45.0 | 40.0±21.6 |
>
> **W3:** Missing algorithmic details. (how to specify $P_\text{DR}, P_\text{Curi}$)
>
> **A:** In AD2S, the sampling ratios $p_\text{DR}$ and $p_\text{Curi}$ govern our two-stage buffer construction process for $\mathcal{D}^\text{re}$: (1) We first calculate density ratio scores on mixed data $\mathcal{D}^\mathrm{off} \cup \mathcal{D}^\mathrm{on}$ using model $w$ and select the top-$k$ transitions with the biggest density ratio scores with ratio $p_\text{DR}$ to build the aligned data buffer $\mathcal{D}^\text{aligned}$, then (2) apply curiosity scores calculated by model $g$ on $\mathcal{D}^\text{aligned}$ to select top-$k$ transitions with biggest curiosity score with ratio of $p_\text{Curi}$ to build the regeneration buffer $\mathcal{D}^\text{re}$.
>
> Although our method introduces additional parameters, it achieves state-of-the-art performance through straightforward hyperparameter tuning via grid search. To validate the effectiveness of our approach, we conducted ablation studies on the Walker2d environment and present the empirical results below. In our experimental settings, we maintain consistent sampling ratios for all tasks within the same benchmark domain (i.e., the locomotion tasks: Walker2d, Hopper, and HalfCheetah). We have added an illustration on how to specify $P_\text{DR}, P_\text{Curi}$ in the method section in the revised paper.
>
> | Data quality | random | medium | medium-replay | medium-expert |
> | -------- | -------- | -------- | ----- | ----- |
> | $p_\text{DR}$=0.1 | 36.8±26.9 | 86.1±1.9 | 120.3±3.4 | 110.7±1.4 |
> | $p_\text{DR}$=0.3 | 91.8±2.5 | 118.6±2.8 | **121.3±3.7** | 123.6±1.7 |
> | $p_\text{DR}$=0.5 | **99.3±11.6** | **119.4±1.2** | 121.0±7.6 | **129.1±4.6** |
> | $p_\text{DR}$=0.7 | 62.4±23.6 | 117.5±1.7 | 119.2±6.7 | 119.8±3.0 |
>
> | Data quality | random | medium | medium-replay | medium-expert |
> | -------- | -------- | -------- | ----- | ----- |
> | $p_\text{Curi}$=0.1 | 78.9±7.6 | 117.7±4.9 | 119.7±2.5 | 124.2±1.2 |
> | $p_\text{Curi}$=0.3 | 77.4±1.6 | 119.4±3.0 | 119.2±5.4 | 124.7±4.1 |
> | $p_\text{Curi}$=0.5 | **99.3±11.6** | **119.4±1.2** | **121.0±7.6** | **129.1±4.6** |
> | $p_\text{Curi}$=0.7 | 81.6±9.8 | 117.8±4.9 | 114.8±2.5 | 123.4±1.8 |
>
> **W4:** Typos
>
> **A:** Thank you for your suggestion. We have corrected the typos in the revised paper.

---

### Author Response · Authors · 2025-12-01
**Rebuttal Summary to AC**

Dear Area Chair,

We sincerely appreciate the reviewers' thoughtful feedback on our work. In this paper, we propose a framework named AD2S that accelerates online fine-tuning in offline-to-online (O2O) RL through diffusion-model-based data generation. Below, we highlight the key questions during rebuttal:

**Limited Experimental Domains:** Reviewers **wo2Y** and **RqZB** raise concerns about the usability of AD2S, claiming that the results are based on limited experimental domains. However, they overlook our experimental results on sparse reward and pixel-based tasks presented in the Appendix. Our comprehensive experiments on these tasks (e.g., AntMaze and V-D4RL) demonstrate the usability of our method, which has already been presented in the Appendix in our original submission.

**Effectiveness of Proposed Advantage:** Reviewers **wo2Y** and **RqZB** think that the advantage proposed in Eq. (8) may not correlate with long-horizon performance in sparse or semi-sparse MDPs. While Eq. (8) uses a short-term reward metric, the comprehensive experimental results in Tables (1) and (4) of our paper demonstrate consistent performance gains across both dense and sparse reward environments. Moreover, we conduct a variant of AD2S that uses policy-dependent advantage $A(s,a) = Q(s,a) - \mathbb{E}_{\hat{a}\sim\pi}[Q(s,\hat{a})]$ and run experiments on sparse reward tasks. Results in the reply to Reviewer **RqZB** on these sparse reward tasks indicate that advantage estimation in Eq. (8) improves the performance on complex tasks while remaining competitive on medium-difficulty tasks. The policy-dependent advantage estimation outputs an unreliable long-term estimate during early fine-tuning, thereby failing to weight the aligned data appropriately. On the contrary, our proposed policy-independent advantage will assign the same weight to states with near-zero reward, and the data alignment process will be dominated by the density ratio or curiosity, thereby continuing to generate high-quality data into the online reply buffer.

**Computational Cost:** Reviewers **wo2Y**, **pLsX**, and **BJwC** raise concerns about the computational cost of AD2S, viewing it as a complex framework. However, AD2S introduces minimal computational overhead, requiring only two additional MLPs on top of the diffusion model. During online fine-tuning, updating two MLP models takes only 0.04 seconds per step, and the Cal-QL algorithm takes 0.3 seconds per step. This represents a marginal increase in computational cost. The ablation studies also demonstrate that our method does not require heavy parameter tuning, and we maintain the same hyperparameters across different tasks within the same domain.

**Dependence on Model Quality:** Reviewer **BJwC** wonders if the performance may degrade with imperfect dynamics or diffusion models. We introduce early stopped diffusion models during online fine-tuning to test the sensitivity of AD2S to imperfect diffusion models. The results presented in the reply to Reviewer **BJwC** indicate that our method continues to improve performance when the diffusion models are not well-trained, demonstrating AD2S's robustness and usability. The overall training process is controllable through appropriate diffusion model training.

**Originality:** Reviewer **pLsX** concerns that AD2S lacks significant technical originality. Compared to current SOTA diffusion-based data generation in O2O RL, EDIS introduces several energy models to guide the diffusion model in generating online distribution data, and CFDG applies a re-weighting trick to balance the offline and online synthetic data during online fine-tuning. However, neither of them aligns offline data to online data distribution for policy training. Moreover, they do not filter high-reward transitions, which encourages the agent to explore the online environment more efficiently. Our method uses the diffusion model to re-generate synthetic data from historical near on-policy transitions and pushes these data towards high-rewarding and underexplored regions. This helps stabilize the early online Q-learning process and encourages the agent to explore the environment and calibrate the offline pretrained model.

Thank you for your consideration.

Best regards,

Authors

---

### Meta-Review · Area_Chair_LX6s · 2025-12-28

**Summary:**

My recommendation for this submission is "Reject". While authors' rebuttal have clarified various points and provided experimental results that resolve reviewers' concerns, major concerns on the significance & impact of the experimental results still remain. I agree with the reviewers in that results are not significant enough to compensate the practical cost of the proposed idea (training diffusion models and generating synthetic data with it).

**Reviewer Concerns:**

Concerns addressed by the rebuttal
- Limited benchmark tasks: Authors have clarified that more results on additional benchmark tasks were available in Appendix A.
- Clarification on advantage estimator: Authors have provided additional analysis on this design choice
- Learning curves: Authors have provided learning curves in the revised draft
- Missing robustness analysis (with different model qualities): Authors provided additional experimental results on this
- Cost analysis: Authors have clarified that additional compute cost is negligible

Outstanding concerns
- Usability and efficiency of the proposed algorithm: In my viewpoint, the rebuttal response is not clearly resolving this concern. Actually, learning curve plots clearly show that the differences between baselines and the proposed algorithm are not statistically significant. This is also noticeable in other results across mujoco, v-d4rl, antmaze & adroit tasks.
- Lack of theoretical grounding: Authors didn't provide a theoretical guarantee for the proposed method. But I don't think this is strictly necessary for all papers.
- Real-world validation or extension to LLMs: Authors didn't provide a new results that can have impact on real-world applications. But I don't think this is strictly necessary.

**Reviewer Scores:**

- Reviewer wo2Y (score 4): I expect this reviewer not to increase the score, considering that requested learning curves are in fact showing that the differences between various methods are not statistically significant.
- Reviewer RqZB (score 2): I expect this reviewer not to increase the score, considering that the reviewer is not satisfied with the learning curve.
- Reviewer BJwC (score 6): I expect this reviewer will remain their positive score, but I don't expect the reviewer to particually champion the paper.
- Reviewer pLsX (score 4): I expect this reviewer to remain their score considering that their concerns on usability, real-world validation still remain.

---

### Decision · Program_Chairs · 2026-01-26

Reject